# Appetite and Nutritional Status as Potential Management Targets in Patients with Heart Failure with Reduced Ejection Fraction—The Relationship between Echocardiographic and Biochemical Parameters and Appetite

**DOI:** 10.3390/jpm11070639

**Published:** 2021-07-06

**Authors:** Marta Kaluzna-Oleksy, Filip Sawczak, Agata Kukfisz, Magdalena Szczechla, Helena Krysztofiak, Marta Wleklik, Katarzyna Przytarska, Jacek Migaj, Magdalena Dudek, Ewa Straburzyńska-Migaj, Izabella Uchmanowicz

**Affiliations:** 11st Department of Cardiology, University of Medical Sciences in Poznan, 61-848 Poznan, Poland; fsawczak@gmail.com (F.S.); agata.kukfisz@gmail.com (A.K.); szczechlamagdalena@gmail.com (M.S.); helenakrysztofiak@gmail.com (H.K.); katarzyna.przytarska@gmail.com (K.P.); protozoan@tlen.pl (J.M.); magdamroz8@gmail.com (M.D.); ewa.straburzynska-migaj@skpp.edu.pl (E.S.-M.); 2Lord’s Transfiguration Clinical Hospital, University of Medical Sciences in Poznan, 61-848 Poznan, Poland; 3Faculty of Health Sciences, Wroclaw Medical University, 50-367 Wroclaw, Poland; marta.wleklik@umed.wroc.pl (M.W.); Izabella.uchmanowicz@umed.wroc.pl (I.U.)

**Keywords:** malnutrition, nutrition-related risk, MNA, BNP

## Abstract

This study aimed to investigate the role of appetite loss and malnutrition in patients with heart failure with reduced ejection fraction (HFrEF). In this prospective, observational, single-center study, we enrolled 120 consecutive adults with HFrEF. We analyzed the selected clinical, echocardiographic, and biochemical parameters. Appetite loss and malnutrition were assessed by CNAQ (Council on Nutrition Appetite Questionnaire) and MNA (Mini Nutritional Assessment)/GNRI (Geriatric Nutritional Risk Index) questionnaires, respectively.Most patients were men (81.7%), mean age was 55.1 ± 11.3 years, and mean left ventricular ejection fraction was 23.9 ± 8.0%. The mean CNAQ score was 28.8 ± 3.9, mean MNA—23.1 ± 2.6, and mean GNRI—113.0 ± 12.3. Based on ROC curves, we showed that a sodium concentration <138 mmol/L had the greatest discriminating power for diagnosing impaired nutritional status (MNA ≤ 23.5) with a sensitivity of 54.5% and specificity of 77.8%. The threshold of HDL <0.97 mmol/L characterized 40.7% sensitivity and 86% specificity, B-type natriuretic peptide >738.6 pg/dL had 48.5% sensitivity and 80.8% specificity, high-sensitivity C-reactive protein >1.8 mg/L had 94.9% sensitivity and 42.9% specificity, and bilirubin >15 µmol/L had 78.2% sensitivity and 56.9% specificity. Nutritional status and appetite assessed by MNA/GNRI and CNAQ questionnaires showed poor correlations with other findings in HFrEF patients.

## 1. Introduction

The number of patients suffering from heart failure (HF), in contrast to other cardiovascular diseases, is systematically growing worldwide. HF affects more than 64 million people worldwide and is referred to as a global epidemic [1]. It is considered a burden not only to patients and their families but also to the society and healthcare systems [2]. Over the next decade, no improvement is expected due to population aging, detrimental lifestyle, exposition to cardiovascular risk factors, and, paradoxically, improving the treatment of acute cardiac conditions. Despite improvements in HF management, mortality remains high [3].

Functional tissue ischemia caused by HF results in the impaired function of kidneys, liver, stomach, and intestines. This may lead to appetite loss, nausea, malabsorption, increased catabolism, weight loss, and in consequence, malnutrition, and cachexia [4,5]. Nausea and the lack of appetite may occur when blood is shifted from the gastrointestinal tract to the more vital organs [6]. Malnutrition, causing weakness of myofibrils, leads to worse heart contractility and decreased cardiac output [4]. Malnutrition is also considered an indicator of the underlying disease and may contribute to its progression or be a prognostic factor [7]. Additionally, low body weight in HF may associate with increased catabolism, which in turn, associate with higher levels of tumor necrosis factor (TNF) and other cytokines, as well as increased cortisol/dehydroepiandrosterone ratio [5]. The lack of appetite and nutritional deficiencies might contribute to cachexia in HF patients [5].

The exact prevalence of malnutrition is difficult to evaluate due to the lack of standardized methods of diagnosis. In hospitalized patients with chronic HF, the prevalence of nutritional risk ranges from 34% to 90%, depending on the applied screening tools and the investigated population [8,9,10,11,12,13].

HF is one of the diseases linked to disease-associated malnutrition (DAM) [14]. In DAM, we observe a higher energy and nutrients supply requirement due to disease-related inflammation, decreased appetite, eating, and swallowing problems. All these factors lead to the risk of body weight loss and might contribute to complications while treating the underlying disease [15]. The SICA-HF study demonstrated that HF patients were at risk of anorexia, which was defined as the lack of the desire to eat and the lack of appetite [16]. Some patients presented cachexia without symptoms of anorexia, and others had anorexia without symptoms of cachexia [17]. According to Andreae et al., patients with heart failure with reduced ejection fraction (HFrEF) and higher NYHA class had worse appetite [18].

Therefore, we aimed to investigate in HFrEF patients the role of appetite loss and malnutrition assessed by CNAQ (Council on Nutrition Appetite Questionnaire) and MNA (Mini Nutritional Assessment)/GNRI (Geriatric Nutritional Risk Index) questionnaires, respectively.

## 2. Materials and Methods

### 2.1. Study Population

In this prospective, observational, single-center study, we enrolled 120 consecutive adults hospitalized between January 2019 and December 2019 due to stable or decompensated chronic HF. The patients were classified according to the International Statistical Classification of Diseases and Related Health Problems (ICD-10) for the diagnosis of HF (I50). The inclusion criteria were: (1) admission due to chronic HF (ICD-10 code for the main diagnosis—I50); (2) age ≥ 18 years; (3) HF history longer than three months; (4) left ventricular ejection fraction (LVEF) < 40%; and (5) signing the informed consent form.

### 2.2. Clinical, Laboratory, and Echocardiographic Data

On admission, a detailed medical history and information on taken drugs were collected. Special attention was paid to other conditions that might have influenced the patient’s nutritional status (e.g., neoplasms) and comorbidities modifying the cardiovascular risk, such as diabetes mellitus (DM), chronic kidney disease (CKD), chronic obstructive pulmonary disease (COPD), arterial hypertension or previous myocardial infarction (MI). Patients were classified according to the New York Heart Association (NYHA) functional classification [19]. Patients underwent physical examination, including measurements of blood pressure (BP), heart rate (HR), height, and body weight. Patients were weighed without shoes and with light clothes on with a standardized and controlled weight with a digital scale and 0.1 kg accuracy. The following formula was used to calculate body mass index (BMI): BMI = weight (kg)/height (m)^2^ [20].

Fasting blood samples were taken. We analyzed complete blood count, B-type natriuretic peptide (BNP), N-terminal pro B-type natriuretic peptide (NT-proBNP), lipid profile, creatinine, fasting glucose, serum albumin, aspartate aminotransferase (AST), alanine aminotransferase (ALT), electrolytes (sodium, potassium), high-sensitivity C-reactive protein (hsCRP), and iron (Fe).

ECG was performed in all patients on admission. Echocardiography was performed in each patient, and LVEF was assessed using Simpson’s rule (according to the guidelines [21]). The appetite and nutritional status were evaluated on admission.

### 2.3. Nutrition Evaluation

The MNA (Mini Nutritional Assessment) questionnaire is a proper tool for assessing malnutrition and a useful tool for assessing high-risk patients with HF [22]. The patient’s nutritional status evaluation was based on the Polish version of the MNA form (provided by Société des Produits Nestlé SA 1994, Revision 2009, Vevy, Switzerland, Trademark Owners, which holds the copyright of the instrument: http://www.mna-elderly.com/, accessed on between January 2019 and December 2019). The MNA questionnaire is a simple, non-invasive tool, first validated over 25 years ago, to assess and demonstrate malnutrition [23,24]. This questionnaire was developed for the evaluation of the elderly [23,24,25,26]. Two versions of this questionnaire are used in clinical practice: a full version developed in 1994 and a short form, known as the MNA-SF version [23,27,28]. Due to clearly defined thresholds clinicians commonly use it in their daily practice all over the world [25]. In the full version of MNA, a score of 24–30 indicates the proper nutritional status, 17–23.5 signifies a risk of malnutrition, and a score below 17 indicates malnutrition [24].

Apart from MNA, we used another instrument to evaluate malnutrition, i.e., the Geriatric Nutritional Risk Index (GNRI). GNRI is a malnutrition assessment tool developed by Bouillanne et al. It can be used as a fast-screening tool [29]. For calculating GNRI, only two variables are needed: serum albumin level and BMI. In our study, GNRI for each patient was derived using the formula (1.489 × serum albumin [g/L]) + (41.7 × body weight/ideal body weight (IBW) [kg]); IBW was calculated as follows: IBW = height^2^ [m] × 22. Patients are categorized into four subgroups: major risk of malnutrition (GNRI < 82), moderate risk (GNRI 82 to <92), low risk (GNRI 92 to ≤98), and no risk (GNRI >98) [29]. For the statistical analysis, we divided patients into two subgroups, GNRI > 98—no risk of malnutrition and GNRI ≤ 98—the risk of malnutrition of any grade.

### 2.4. Appetite Evaluation

To assess the appetite, both CNAQ (Council on Nutrition Appetite Questionnaire) and its shorter version—SNAQ (Simplified Nutritional Appetite Questionnaire) can be used. CNAQ questionnaire can be used to assess patients before and during the development of malnutrition and to observe any changes of appetite in time. CNAQ and SNAQ demonstrated sound psychometric properties and can be used to measure appetite in patients with HF [30]. Even though SNAQ is shorter than CNAQ, it is recommended, where possible, to use the full CNAQ because of its greater reliability [30]. A threshold ≤28 points indicates poor appetite and classifies the patient as being at risk of significant weight loss within six months [31]. We used a Polish version of the CNAQ questionnaire that was adapted to assess Polish patients with HF as well as those with reduced as well as preserved ejection fraction [28].

### 2.5. Statistical Analysis

Received values were presented as mean ± SD for continuous variables and N (%) for categorical ones. After analyzing MNA, CNAQ, and GNRI results, the study cohort was divided into groups to conduct statistical analysis. It was done with the use of CNAQ to assess appetite. The cohort consisted of subgroups with appetite dysfunction (defined as CNAQ score ≤ 28) and no impairment of appetite (CNAQ score > 28). Based on MNA score, one group presented proper nutritional status (MNA score > 23.5), and the other group comprised of patients at risk of malnutrition and malnourished (MNA score ≤ 23.5). Based on GNRI, one group presented with no nutrition-related risk (GNRI > 98), and the other group characterized a nutrition-related risk (GNRI ≤ 98).

Several clinical, biochemical, and echocardiographic parameters were analyzed in each group and compared using the U-Mann–Whitney test for continuous variables. Chi2 test, with Yates correction when needed, was used to compare categorical variables. Spearman’s nonparametric correlation was performed to assess the relationship between MNA score, CNAQ score, and GNRI score with continuous variables and each other. The receiver operating characteristic (ROC) curve was used to determine the cut-off points of BNP and other continuous variables for predicting impaired nutritional status (MNA < 23.5). We selected cut-off points with the maximum Youden index and assessed specificity, sensitivity, positive predictive value (PPV), and negative predictive value (NPV), as well as area under the curve (AUC). We considered *p* < 0.05 as statistically significant. All statistics were performed using Statistica version 13.3 software (Statsoft, Tulsa, OK, USA).

## 3. Results

### 3.1. Baseline Study Population Characteristics

The study sample consisted of 120 patients with HFrEF, and 98 (81.7%) were men. The mean age was 55.1 ± 11.3 years (median 57 years). A total of 59 (49.2%) of patients had NYHA class I or II, 50 (41.7%)—NYHA class III, and 11 (9.2%)—NYHA class IV. A total of 39 (32.5%) patients were hospitalized due to chronic HF exacerbation. The mean LVEF was 23.9 ± 8.0% (median 25%). At the time of inclusion, 116 (96.7%) patients were treated with beta-blockers, 89 (74.2%) with angiotensin-converting enzyme inhibitors or with angiotensin receptor blockers, and 104 (86.7%) with mineralocorticoid receptor antagonists. 20 (16.7%) patients were discharged on sacubitril/valsartan (initiated after collecting all required parameters). The most common comorbidities were hypertension—64 cases (53.3%), ischemic heart disease—56 (46.7%), and diabetes—36 (30%) (Table 1).

### 3.2. Nutrition Assessment

The mean CNAQ score was 28.8 ± 3.9, mean MNA—23.1 ± 2.6, and mean GNRI—113.0 ± 12.3. According to the CNAQ questionnaire, 50 (41.7%) patients had impaired appetite. Based on the MNA questionnaire, 66 (55%) patients were at risk of malnutrition or malnourished (MNA score ≤ 23.5). We calculated GNRI in 97 of 120 (23 lacked serum albumin level data). Based on GNRI, 88 (90.7%) patients revealed no nutrition-related risk (GNRI > 98), 5 (5.2%) patients presented a low nutrition-related risk (GNRI: 92 to ≤98), 4 (4.1%) patients had a moderate nutrition-related risk (GNRI: 82 to <92), and no patients had a major nutrition-related risk (GNRI: <82) (Table 1).

Comparing patients with impaired vs. normal appetite showed no significant differences in biochemical and echocardiographic parameters and CNAQ score (Table 2).

We observed no significant differences in the MNA score between the group with normal nutritional status and the group comprised of patients at risk of malnutrition and with stated malnutrition (Table 3). Analyzing clinical parameters, heart rate on discharge was higher in patients with the impaired nutritional status. We also observed significantly lower sodium concentration, total serum protein, total cholesterol, and HDL cholesterol levels in this group. The population with the inappropriate nutritional status also exhibited higher BNP, GGTP, bilirubin, and hsCRP concentrations. Echocardiographic as well as other analyzed clinical and biochemical parameters revealed no significant differences between groups (Table 3).

Patients with the nutrition-related risk (GNRI ≤ 98) had significantly lower systolic blood pressure on admission than those with no nutrition-related risk. The MNA score was lower in this group and among echocardiographic parameters the left ventricle end-diastolic diameter (LVEDD) was also lower. No differences were reported for other variables, aside from albumin level and BMI, but they were part of the GNRI formula (Table 4).

### 3.3. Correlation Analysis

Furthermore, we correlated chosen continuous clinical, biochemical, and echocardiographic parameters with CNAQ, MNA, and GNRI scores. We observed significant positive correlations between MNA score and CNAQ (r = 0.28; *p* = 0.002), sodium concentration (r = 0.27, *p* = 0.003), as well as HDL cholesterol level (r = 0.24, *p* = 0.015). We also showed a significant negative correlation between MNA score and BNP (r = −0.23, *p* = 0.011) and hsCRP levels (r = −0.23, *p* = 0.023). There was a significant correlation between total protein level and MNA (r = 0.25, *p* = 0.013); however, no correlation between MNA and albumin level was observed. No significant correlations between CNAQ score and age, eGFR, GNRI, BMI, or any other continuous parameter were observed either.

Additional analysis was made to assess correlations between biochemical parameters and GNRI score. We revealed a positive correlation between age (r = −0.21, *p* = 0.04), systolic blood pressure (r = 0.35, *p* = 0.0006) and diastolic blood pressure (r = 0.43, *p* = 0.00002) on admission, hemoglobin level (r = 0.40, *p* = 0.00006), total serum protein (r = 0.25, *p* = 0.02) triglycerides concentration (r = 0.37, *p* = 0.0003), and LDL level (r = 0.26, *p* = 0.02). From echocardiography parameters, only LVEDD positively correlated with GNRI (r = 0.39, *p* = 0.001). Parts of GNRI formula—albumin level (r = 0.51, *p* < 0.00001) and BMI (r = 0.78, *p <* 0.00001) also correlated with its score. A negative correlation was obtained between GNRI score and both BNP (r = −0.30, *p* = 0.003) and HDL levels (r = −0.22, *p* = 0.03). Parts of GNRI formula—albumin (r = 0.51, *p <* 0.00001 and BMI (r = 0.78, *p <* 0.00001) also correlated with its score but association with them is apparent. GNRI score was not associated with MNA nor CNAQ score or any other variables.

### 3.4. ROC Curve Analysis

The best AUC characterized ROC curves of sodium (AUC = 0.699; 95% CI 0.606–0.792; *p* < 0.001), HDL (AUC = 0.654; 95% CI 0.550–0.758; *p* = 0.004) and BNP (AUC = 0.646; 95% CI 0.546–0.746; *p* = 0.004) (Figure 1) levels. The ROC curve assessment revealed that sodium concentration < 138 mmol/L had the greatest discriminating power for the impairment of nutritional status in MNA (MNA ≤ 23.5) with sensitivity of 54.5% and specificity of 77.8%. HDL concentration < 0.97 mmol/L provided 40.7% sensitivity and 86% specificity for nutritional problems. BNP concentration > 738.6 pg/dL predicted impairment of nutritional status with sensitivity of 48.5% and specificity of 80.8%. hsCRP> 1.8 mg/L resulted in prediction of nutrition impairment with 94.9% sensitivity and 42.9% specificity. Concentration of bilirubin > 15 µmol/L allowed prediction of MNA score ≤ 23.5 with sensitivity of 78.2% and specificity of 56.9%, which resulted in the highest Youden Index for that cut-off point (0.35). AUC of ROC curves for other analyzed parameters and selected cut-off values with sensitivity, specificity, PPV, and NPV are presented in Figure 1.

## 4. Discussion

The issue of nutrition is rarely discussed or evaluated in a health care setting. Study data revealing a complex relationship between appetite and clinical, biochemical, and echocardiographic parameters in HFrEF patients is not available in the literature—our study appears to be the first one.

It is challenging to evaluate patients with the impaired appetite and the abnormal nutritional status due to a lack of standardized diagnostic methods. Yet, early detection of poor appetite in HF patients seems to be crucial in preventing malnutrition. Reduced desire to eat and related malnutrition may contribute to the deterioration of cardiac cachexia and consequently worsen the prognosis [32,33]. Even 90% of HF patients may experience malnutrition [8,9,10,11,12,13], which is considered a factor of patients’ life quality [14,31].

In our study, impaired appetite was observed in 41.7% of patients. Andreae et al. disclosed a similar frequency of loss of appetite (38.0% vs. 41.7%, respectively) [34]. However, the population analyzed in that study was not as homogenous as in ours. As inclusion criteria, they mentioned HF symptoms and mild to severe left ventricular systolic dysfunction; therefore, LVEF was not strictly specified. In contrast to our study, where only hospitalized patients were enrolled, they recruited subjects in the outpatient setting. Interestingly, Song et al. showed that poor appetite was presented in more than 60% of HF patients [35]. A significant difference in poor appetite rates can be related to the analyzed population. Song et al. enrolled only patients hospitalized due to HF exacerbation, while we mainly included patients with stable HF (only 31.6% had symptoms of decompensation). What is more, Song et al. evaluated appetite as an element of many symptoms listed in the Memorial Symptom Assessment Scale Heart Failure (MSAS-HF), where patients were asked about the presence of the lack of appetite, its frequency, severity, and how inconvenienced they were by it. We, however, assessed patients with a standardized questionnaire, i.e., CNAQ. Additionally, the analyzed population was not as homogenous as in our study. They included HF patients with both reduced left ventricular systolic function (LVEF < 40%) or preserved systolic function (LVEF ≥ 40%) [35]. Additionally, Landi et al. revealed decreased food intake or poor appetite in 65.1% (339 out of 521) of HF patients [36]. Nonetheless, in our study, we considered many more components of appetite using the CNAQ score than Landi et al., where the definition of malnutrition was stated using only two questions.

We did not observe any associations between decreased appetite and higher NYHA class, reported by Andreae et al. [34]. Our patients mainly had NYHA class II (45%), similarly to Andreae et al. (61%); however, our population was younger (median age: 57 years vs. 72 years). What seems to be important, Andreae et al. did not differentiate between HFrEF and HFpEF. Analogically, in our research, age did not correlate with appetite. Contrarily, Andreae et al. showed that older age was related to lower appetite [34].

According to Landi et al., anorexia, defined as a presence of decreased food intake or an occurrence of poor appetite, also seems to be associated with age [36]. The mean age of Landi’s population was 80.4 ± 7.5 years; meanwhile, it was only 55.1 ± 11.3 years in our study. As distinct from our research, in studies by Landi et al. and Andreae et al., authors proved the association of decreased appetite and age, but recruited an older population than ours [34,36]. The difference may be explained by the pathophysiological process of hormonal changes during aging. It was shown that the appetite-related hormone, ghrelin, was reduced in older people, and this may have adverse effects on gastric emptying and intestine motility [37,38].

Our research showed no significant association between appetite and gender. Similarly, Andreae et al. revealed no association between gender and CNAQ score [34]. Oppositely, Landi et al. found a relation between impaired appetite and female gender [36]. However, the percentage of women in our group was much lower than in the study by Landi et al. (18.3% vs. 60%, respectively).

The correlations between biochemical parameters and CNAQ score have not yet been evaluated in patients with HFrEF. In our research, we observed no association between serum albumin level and CNAQ. Arshad et al. showed a correlation between serum albumin level and CNAQ, which can be used to predict hypoalbuminemia and malnutrition [39]. However, the authors included patients suffering from end-stage chronic renal disease, not HF. Patients enrolled by Arshad et al. and population in our study seemed to be similar considering the mean age (53.5 ± 14.4 vs. 55.1 ± 11.3 years, respectively), a percentage of men (83.6% vs. 81.7%, respectively), and mean serum albumin level (38.5 ± 7.6 g/L vs. 40.3 ± 5.1 g/L, respectively) [39].

In our study, the number of patients with HFrEF and malnutrition or being at risk of malnutrition according to MNA score was smaller than in other studies using MNA to assess the nutritional status [9,10,11,13]. In our study, 55% of patients had the impaired nutritional status, i.e., were at risk of malnutrition or malnourished. Bonilla-Palomas et al. revealed 72.6%; however, the analyzed group was older (mean age 55.1 ± 11.3 vs. 73.0 ± 10.1 years) [9]. In the study by Sargento et al., the population consisted of 6% of malnourished patients and 10% of patients were at risk of malnutrition [40]. The possible reason for a better nutritional status in Sargento et al. study was enrolling only outpatient subjects, not hospitalized during the last three months compared to our hospitalized population. Yost et al. reported that 24.7% of patients were malnourished and 47.5% were at risk of malnutrition, and only 9.9% were classified as well-nourished [11]. This significantly higher occurrence of the impaired nutritional status was probably because of more advanced HF in their population. The assessment was performed in a preparation period for left ventricular assist device (LVAD) implantation or heart transplantation process. In the study of Aggarwal et al., 22.1% of patients were malnourished, and 68.2% were at risk of malnutrition with just 9.7% having proper nutritional status [10]. The population in this research was similar as in the study by Yost et al. assessing subjects with NYHA class III/IV in a preparation period for LVAD implantation or heart transplantation process or decompensated HF [11].

We observed a higher heart rate on admission in the group consisting of malnourished patients and at risk of malnutrition compared with a well-nourished group. This finding was proven by another study [13]. We also observed statistically higher BNP levels in patients with the inappropriate nutritional status, whereas Aggarwal et al. did not show any association between the nutritional status and natriuretic peptides [10]. In the population of Aggarval et al., all subjects were in NYHA class III/IV. In contrast, in our study NYHA class II was the most common class. In our work, patients with the impaired nutritional status (malnourished and at risk of malnutrition) had higher mean GGTP (89.4 ± 99.9 vs. 126.5 ± 126.1 U/L), mean bilirubin level (17.0 ± 9.1 vs. 21.3 ± 11.9 µmol/L), and mean hsCRP concentration (mean 5.3 ± 6.8 vs. 10.2 ± 15.3 mg/L) compared to those with the normal nutritional status. Kaluzna-Oleksy et al. did not reveal these associations in a group of similar age (mean 55 ± 11 years vs. 55.1 ± 11.3 years), but higher NYHA class (NYHA III and IV: 50.8% vs. 62.8%) [13]. In our study the correlation between CNAQ and MNA scores was significant (r = 0.28, *p <* 0.001) but not as strong as in Wleklik et al. study (r = 0.8, *p* < 0.001) [28].

According to our results, 9.3% of patients were malnourished, if measured with the GNRI form (GNRI ≤ 98), while in the study of Sze et al., the occurrence of malnutrition was at the level of 19% (also assessed with the GNRI form) [41]. Sze et al. analyzed patients with HFrEF and HFpEF, in contrast to our more homogeneous population of HFrEF patients. What is more, our patients were, on average, younger (median age was 57 years vs. 76 years, respectively). Another research by Sze et al. revealed 13% of patients with GNRI ≤ 98 where the group was also not homogenous and included patients with no pre-existing HF [42]. Additionally, the study population was older than ours (median age was 57 years vs. 75 years, respectively).

Considering echocardiographic parameters, in our work, a mean LVEDD was lower in patients with GNRI ≤ 98. Yasumura et al. reported no difference between groups with GNRI ≤ 92 and GNRI > 92 [43]. The reason for such dissimilarity could have been a different cut-off point of GNRI score, a significantly higher LVEF of Yasumura et al. patients (mean 50 ± 17% vs. 23.9 ± 8.0%), and enrolment of patients with HF exacerbation.

We also analyzed correlations of GNRI score with continuous parameters mentioned in Table 1. In our study GNRI correlated to an older age. The same was observed in Sze et al. and Yoshihisa et al. studies [44,45]. We obtained a significant correlation between GNRI and systolic as well as diastolic blood pressure. Sze et al. [44] revealed the same result, although in the study of Yoshihisa et al. [45], the association between GNRI and systolic and diastolic blood pressure was not stated.

In our study, hemoglobin level and total serum protein level positively correlated with GNRI and BNP and negatively correlated with GNRI. Identically Yoshihisa et al. proved all these findings [45]. Conterminously, Sze et al. proved the correlation with hemoglobin level, but total serum protein level and BNP were not analyzed [44].

Presented research shows some limitations. This was a single-center study and included patients with stable as well as exacerbated HF.

## 5. Conclusions

Nutritional status assessed by MNA/GNRI and appetite assessed by CNAQ questionnaires showed poor correlations with other findings in HFrEF patients. It was possible to determine cut-off points for BNP to identify patients at risk of malnutrition, but the clinical significance of this finding requires further investigation. Nutritional status as assessed by MNA/GNRI questionnaires did not correlate significantly with an appetite as assessed by the CNAQ questionnaire in HFrEF patients.

## Figures and Tables

**Figure 1 jpm-11-00639-f001:**
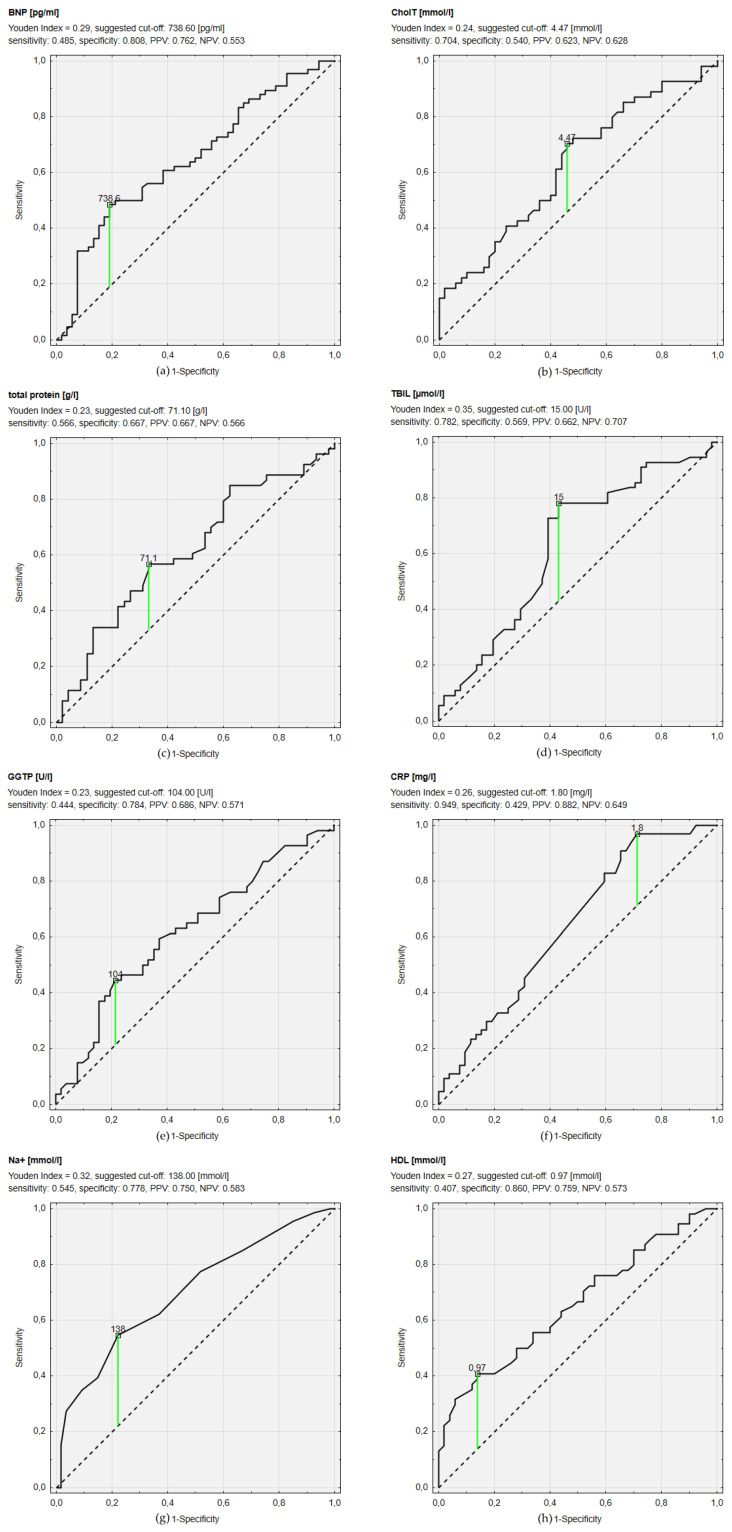
Receiver operating characteristic curves of different biochemical parameters concentrations predicting impaired nutritional status (defined as The Mini Nutritional Assessment (MNA) ≤ 23.5). (**a**): Receiver operating characteristic curves of B-type natriuretic peptide (BNP); (**b**): Receiver operating characteristic curves of total cholesterol (CholT); (**c**): Receiver operating characteristic curves of total protein; (**d**): Receiver operating characteristic curves of total bilirubin (TBIL); (**e**): Receiver operating characteristic curves of gamma-glutamyltranspeptidase (GGTP); (**f**): Receiver operating characteristic curves of C-reactive protein (CRP); (**g**): Receiver operating characteristic curves of sodium concentration (Na+); (**h**): Receiver operating characteristic curves of high density lipoprotein (HDL).PPV—positive predictive value; NPV—negative predictive value.

**Table 1 jpm-11-00639-t001:** Baseline characteristics.

Characteristics	Value ± SD
age (years)	55.1 ± 11.3
men	98 (81.7%)
BMI (kg/m^2^)	28.8 ± 5.6
IHD etiology	56 (46.7%)
HF exacerbation	39 (32.5%)
SBP on admission (mmHg)	112.4 ± 22.1
DBP on admission (mmHg)	73.4 ± 13.2
HR on discharge (beats per minute)	74.7 ± 13.0
Comorbidities	N (%)
DM	36 (30.0%)
COPD	10 (8.3%)
CKD	21 (17.5%)
hypertension	64 (53.3%)
NYHA class	N (%)
I	5 (4.2%)
II	54 (45.0%)
III	50 (41.7%)
IV	11 (9.2%)
I–II	59 (49.2%)
III–IV	61 (50.8%)
Biochemical parameters	Value ± SD
Na^+^ (mmol/L)	138.8 ± 3.7
BNP (pg/mL)	678.3 ± 690.4
NT-proBNP (pg/mL)	3689.2 ± 5133.0
uric acid (µmol/L)	464.2 ± 135.8
creatinine (µmol/L)	105.0 ± 35.7
eGFR (mL/min)	72.6 ± 25.2
K^+^ (mmol/L)	4.28 ± 0.41
hsCRP (mg/L)	8.0 ± 12.5
fasting glucose (mmol/L)	6.40 ± 1.86
serum protein (g/L)	71.1 ± 7.4
serum albumin (g/L)	40.3 ± 5.1
TBIL (µmol/L)	19.3 ± 10.8
ALT (U/L)	37.1 ± 22.2
GGTP (U/L)	108.5 ± 115.2
CholT (mmol/L)	4.32 ± 1.11
TG (mmol/L)	1.60 ± 0.88
LDL (mmol/L)	2.60 ± 0.90
HDL (mmol/L)	1.20 ± 0.32
Hb (mmol/L)	8.92 ± 1.12
Nutritional parameters	
CNAQ score	28.8 ± 3.9
impaired appetite CNAQ ≤ 28	50 (41.7%)
normal appetite CNAQ > 28	70 (58.3%)
MNA score	23.1 ± 2.6
malnourished (MNA score < 17)	1 (0.8%)
at risk of malnutrition (MNA score 17–23.5)	65 (54.2%)
normal nutritional status (MNA score > 23.5)	54 (45.0%)
GNRI score	113.0 ± 12.3
GNRI < 82	0 (0%)
GNRI ≥ 82 and <92	4 (4.1%)
GNRI ≥ 92 and ≤98	5 (5.2%)
GNRI > 98	88 (90.7%)
Medications	
loop diuretics	111 (92.5%)
thiazides	21 (17.5%)
ß-blocker	116 (96.7%)
ACEI/ARB	89 (74.2%)
ARNI	20 (16.7%)
MRA	104 (86.7%)
Ca-blocker	9 (7.5%)
statin	73 (60.8%)
Echocardiographic parameters	
LVEF (%)	23.9 ± 8.0
LVEDD (mm)	70.2 ± 10.8
RVD (mm)	37.4 ± 8.0
LAD (mm)	52.8 ± 10.8
IVS (mm)	10.1 ± 1.8
PWD (mm)	10.2 ± 1.8
Ao (mm)	33.6 ± 4.9

BMI—body mass index, IHD—ischemic heart disease, SBP—systolic blood pressure, DBP—diastolic blood pressure, HR—heart rate, DM—diabetes mellitus, COPD—chronic obstructive pulmonary disease, CKD—chronic kidney disease, NYHA—New York Heart Association Classification, Na^+^ sodium concentration, BNP—B-type natriuretic peptide, NT-proBNP—N-terminal pro B-type natriuretic peptide, eGFR—estimated glomerular filtration rate, K^+^—potassium concentration, hsCRP-high-sensitivity C-reactive protein, TBIL—total bilirubin, ALT—alanine transaminase, GGTP—gamma-glutamyltranspeptidase, CholT—total cholesterol, TG—triglycerides, LDL—low density lipoprotein, HDL—high density lipoprotein, Hb—hemoglobin, CNAQ—The Council on Nutrition Appetite Questionnaire, MNA—The Mini Nutritional Assessment, GNRI—Geriatric Nutritional Risk Index, ACEI—angiotensin converting enzyme inhibitor, ARB—angiotensin receptor blocker, ARNI—angiotensin receptor-neprilysin inhibitor, MRA—mineralocorticoid receptor antagonist, LVEF—left ventricular ejection fraction, LVEDD—left ventricular end-diastolic diameter, RVD—right ventricular diameter, LAD—left atrium diameter, IVS—interventricular septum thickness, PWD—posterior wall of left ventricle, Ao—aorta.

**Table 2 jpm-11-00639-t002:** Differences in chosen parameters in patients with normal vs. impaired appetite based on CNAQ score.

Characteristics	Good Appetite, CNAQ Score > 28	Impaired Appetite, CNAQ Score ≤ 28	*p*
	(*n* = 70)	(*n* = 50)	
age (years)	54.4 ± 11.0	56.2 ± 11.7	0.14
men	59 (84.3%)	39 (78.0%)	0.38
BMI (kg/m^2^)	28.5 ± 5.8	29.3 ± 5.4	0.45
IHD etiology	35 (50.7%)	20 (40%)	0.25
HF exacerbation	20 (29.4%)	17 (34.0%)	0.60
SBP on admission (mmHg)	111.6 ± 23.8	113.3 ± 19.8	0.66
DBP on admission (mmHg)	72.4 ± 13.2	74.8 ± 13.2	0.38
HR on discharge (beats per minute)	74.9 ± 13.0	74.3 ± 13.2	0.94
Comorbidities			
DM	19 (27.1%)	16 (32.0%)	0.60
COPD	5 (7.1%)	5 (10.0%)	0.82
CKD	13 (18.6%)	7 (14.3%)	0.71
hypertension	35 (50.0%)	29 (58.0%)	0.39
NYHA class			
I	4 (5.7%)	1 (2.0%)	0.32
II	33 (47.1%)	21 (42.0%)	0.58
III	28 (40.0%)	22 (44.0%)	0.30
IV	5 (7.1%)	6 (12.0%)	0.36
I–II	37 (52.9%)	22 (44.0%)	0.34
III–IV	33 (47.1%)	28 (56.0%)	
Biochemical parameters			
Na^+^ (mmol/L)	138.9 ± 3.5	138.7 ± 3.9	0.78
BNP (pg/mL)	695.5 ± 729.8	654.1 ± 637.4	0.84
NT-proBNP (pg/mL)	4030.4 ± 5913.7	3158.7 ± 3646.7	0.45
uric acid (µmol/L)	456.8 ± 123.6	472.9 ± 149.8	0.72
creatinine (µmol/L)	104.8 ± 37.1	105.1 ± 34.0	0.75
eGFR (ml/min)	74.1 ± 26.8	70.5 ± 23.1	0.52
K^+^ (mmol/L)	4.31 ± 0.38	4.25 ± 0.44	0.60
hsCRP (mg/L)	9.7 ± 14.9	5.5 ± 7.1	0.09
fasting glucose (mmol/L)	6.45 ± 1.98	6.32 ± 1.67	0.95
serum protein (g/L)	71.7 ± 6.7	70.4 ± 8.2	0.64
serum albumin (g/L)	40.6 ± 4.2	40.1 ± 6.0	0.44
TBIL (µmol/L)	17.8 ± 8.8	21.0 ± 12.8	0.24
ALT (U/L)	37.3 ± 21.7	36.8 ± 23.2	0.89
GGTP (U/L)	98.4 ± 96.1	120.5 ± 134.4	0.92
CholT (mmol/L)	4.28 ± 0.99	4.37 ± 1.26	0.69
TG (mmol/L)	1.52 ± 0.90	1.70 ± 0.85	0.12
LDL (mmol/L)	2.55 ± 0.85	2.67 ± 0.96	0.63
HDL (mmol/L)	1.22 ± 0.31	1.16 ± 0.33	0.35
Hb (mmol/L)	8.88 ± 1.14	8.98 ± 1.10	0.50
Nutritional parameters			
CNAQ score	31.3 ± 2.0	25.2 ± 3.1	*p* < 0.001
MNA score	23.6 ± 2.3	22.5 ± 2.8	0.04
malnourished (MNA score < 17)	0	1 (2.0%)	0.23
at risk of malnutrition (MNA score 17–23.5)	36 (51.4%)	18 (36.0%)	0.09
normal nutritional status (MNA score > 23.5)	34 (48.6%)	31 (62.0%)	0.15
GNRI	112.5 ± 11.8	113.6 ± 12.9	0.66
Medications			
Loop diuretics	65 (94.2%)	46 (92.0%)	0.63
thiazides	8 (11.6%)	12 (24.0%)	0.12
ß-blocker	69 (98.6%)	48 (96.0%)	0.78
ACEI/ARB	51 (73.9%)	38 (76.0%)	0.80
ARNI	11 (16.2%)	7 (14.0%)	0.95
MRA	57 (82.6%)	47 (94.0%)	0.12
Ca-blocker	6 (8.7%)	2 (4.0%)	0.52
statin	42 (60.9%)	31 (62.0%)	0.90
Echocardiographic parameters			
LVEF (%)	23.3 ± 7.4	24.7 ± 8.8	0.36
LVEDD (mm)	70.1 ± 10.4	70.3 ± 11.5	0.69
RVD (mm)	37.1 ± 7.8	37.7 ± 8.2	0.37
LAD (mm)	53.6 ± 11.5	51.5 ± 9.7	0.56
IVS (mm)	10.1 ± 1.9	10.3 ± 1.9	0.72
PWD (mm)	10.0 ± 1.5	10.4 ± 2.1	0.33
Aorta (mm)	33.7 ± 4.0	33.4 ± 5.9	0.92

BMI—body mass index, IHD—ischemic heart disease, SBP—systolic blood pressure, DBP—diastolic blood pressure, HR—heart rate, DM—diabetes mellitus, COPD—chronic obstructive pulmonary disease, CKD—chronic kidney disease, NYHA—New York Heart Association Classification, Na^+^ sodium concentration, BNP—B-type natriuretic peptide, NT-proBNP—N-terminal pro B-type natriuretic peptide, eGFR–estimated glomerular filtration rate, K^+^—potassium concentration, hsCRP-high-sensitivity C-reactive protein, TBIL—total bilirubin, ALT—alanine transaminase, GGTP—gamma-glutamyltranspeptidase, CholT—total cholesterol, TG—triglycerides, LDL—low density lipoprotein, HDL—high density lipoprotein, Hb—hemoglobin, CNAQ—The Council on Nutrition Appetite Questionnaire, MNA—The Mini Nutritional Assessment, GNRI—Geriatric Nutritional Risk Index, ACEI—angiotensin converting enzyme inhibitor, ARB—angiotensin receptor blocker, ARNI—angiotensin receptor-neprilysin inhibitor, MRA—mineralocorticoid receptor antagonist, LVEF—left ventricular ejection fraction, LVEDD—left ventricular end-diastolic diameter, RVD—right ventricular diameter, LAD—left atrium diameter, IVS—interventricular septum thickness, PWD—posterior wall of left ventricle, Ao—aorta.

**Table 3 jpm-11-00639-t003:** Comparison of patients with normal and impaired nutritional status based on MNA questionnaire.

Characteristics	Normal Nutritional Status (MNA > 23.5) (*n*= 54)	At Risk of Malnutrition + Malnutrition (MNA ≤ 23.5) (*n* = 66)	*p*
age (years)	55.1 ±10.1	55.2 ± 12.3	0.73
men	44 (81.5%)	54 (81.8%)	0.85
BMI (kg/m^2^)	29.5 ± 5.4	28.3 ± 5.8	0.23
IHD etiology	23 (42.6%)	32 (48.4%)	0.47
HF exacerbation	14 (25.9%)	23 (34.8%)	0.30
SBP on admission (mmHg)	115.8 ± 24.9	109.5 ± 19.1	0.09
DBP on admission (mmHg)	75.9 ± 12.7	71.4 ± 13.3	0.09
HR on discharge (beats per minute)	71.3 ± 11.1	77.5 ± 13.8	0.017
Comorbidities			
DM	16 (29.6%)	19 (28.8%)	0.87
COPD	3 (5.6%)	7 (10.6%)	0.50
CKD	10 (18.5%)	10 (15.2%)	0.83
hypertension	27 (50.0%)	37 (56.1%)	0.51
NYHA class			
I	2 (3.7%)	3 (4.5%)	0.82
II	25 (46.3%)	29 (44.0%)	0.80
III	24 (44.4%)	26 (39.4%)	0.58
IV	3 (5.6%)	8 (12.1%)	0.21
I–II	27 (50%)	34 (51.5%)	0.87
III–IV	27 (50%)	32 (48.5%)	
Biochemical parameters			
Na^+^ (mmol/L)	140.1 ± 3.2	137.7 ± 3.7	0.0001
BNP level (pg/mL)	559.6 ± 731.6	771.7 ± 646.4	0.006
NT-proBNP (pg/mL)	3618.4 ± 6324.6	3750 ± 3915.0	0.07
uric acid (µmol/L)	445.8 ± 122.4	481.5 ± 481.5	0.18
creatinine (µmol/L)	108.0 ± 34.5	102.5 ± 36.7	0.26
eGFR	69.8 ± 24.2	74.9 ± 26.0	0.26
K^+^ (mmol/L)	4.32 ± 0.35	4.26 ± 0.46	0.43
hsCRP (mg/L)	5.3 ± 6.8	10.2 ± 15.3	0.01
fasting glucose (mmol/L)	6.22 ± 1.61	6.55 ± 2.04	0.31
serum protein (g/L)	72.5 ± 7.0	70.0 ± 7.6	0.049
serum albumin (g/L)	40.6 ± 4.5	40.1 ± 5.7	0.60
bilirubin (µmol/L)	17.0 ± 9.1	21.3 ± 11.9	0.022
ALT (U/L)	38.1 ± 20.3	36.3 ± 23.9	0.26
GGTP (U/L)	89.4 ± 99.9	126.5 ± 126.1	0.037
CholT (mmol/L)	4.58 ± 1.08	4.08 ± 1.09	0.026
TG (mmol/L)	1.68 ± 0.90	1.52 ± 0.85	0.28
LDL (mmol/L)	2.73 ± 0.88	2.48 ± 0.90	0.12
HDL (mmol/L)	1.28 ± 0.31	1.11 ± 0.30	0.006
Hb (mmol/L)	8.92 ± 1.00	8.92 ± 1.22	0.59
Nutritional parameters			
CNAQ score	29.9 ± 3.2	27.8 ± 4.2	0.012
MNA score	25.5 ± 0.9	21.1 ± 1.6	*p* < 0.001
impaired appetite CNAQ ≤ 28	18 (33.3%)	32 (48.5%)	0.09
GNRI	114.3 ± 11.2	111.7 ± 13.2	0.29
Medications			
loop diuretics	51 (94.4%)	60 (90.9%)	0.43
thiazides	6 (11.1%)	14 (21.1%)	0.23
beta-blocker	54 (100%)	63 (95.4%)	0.32
ACEI/ARB	40 (74.0%)	49 (74.2%)	0.88
ARNI	10 (18.5%)	8 (12.3%)	0.47
MRA	48 (88.8%)	57 (86.4%)	0.92
Ca-blocker	3 (5.6%)	5 (7.6%)	0.68
statin	35 (64.8%)	38 (57.6%)	0.35
Echocardiographic parameters			
LVEF (%)	23.8 ± 7.7	24.0 ± 8.3	0.93
LVEDD (mm)	71.9 ± 10.1	68.8 ± 11.2	0.14
RVD (mm)	37.8 ± 9.9	37.0 ± 5.9	0.76
LAD (mm)	53.2 ± 11.1	52.4 ± 10.7	0.88
IVS (mm)	10.3 ± 1.7	10.1 ± 2.0	0.43
PWD (mm)	10.2 ± 1.4	10.1 ± 2.0	0.51
Ao (mm)	34.5 ± 3.9	32.8 ± 5.5	0.07

BMI—body mass index, IHD—ischemic heart disease, SBP—systolic blood pressure, DBP—diastolic blood pressure, HR—heart rate, DM—diabetes mellitus, COPD—chronic obstructive pulmonary disease, CKD—chronic kidney disease, NYHA—New York Heart Association Classification, Na^+^ sodium concentration, BNP—B-type natriuretic peptide, NT-proBNP—N-terminal pro B-type natriuretic peptide, eGFR–estimated glomerular filtration rate, K^+^—potassium concentration, hsCRP-high-sensitivity C-reactive protein, TBIL—total bilirubin, ALT—alanine transaminase, GGTP—gamma-glutamyltranspeptidase, CholT—total cholesterol, TG—triglycerides, LDL—low density lipoprotein, HDL—high density lipoprotein, Hb—hemoglobin, CNAQ—The Council on Nutrition Appetite Questionnaire, MNA—The Mini Nutritional Assessment, GNRI—Geriatric Nutritional Risk Index, ACEI—angiotensin converting enzyme inhibitor, ARB—angiotensin receptor blocker, ARNI—angiotensin receptor-neprilysin inhibitor, MRA—mineralocorticoid receptor antagonist, LVEF—left ventricular ejection fraction, LVEDD—left ventricular end-diastolic diameter, RVD—right ventricular diameter, LAD—left atrium diameter, IVS—interventricular septum thickness, PWD—posterior wall of left ventricle, Ao—aorta.

**Table 4 jpm-11-00639-t004:** Comparison of patients with nutrition-related risk present and no nutrition-related risk based on GNRI formula.

Characteristics	No Nutrition-Related Risk (GNRI > 98)	Nutrition-Related Risk Present (GNRI ≤ 98)	*p*
	(*n* = 88)	(*n* = 9)	
age (years)	53.9 ± 10.9	57.3 ± 8.0	0.44
men	71 (80.7%)	6 (66.7%)	0.58
BMI (kg/m^2^)	28.9 ± 5.5	22.8 ± 2.6	0.0001
IHD etiology	41 (46.6%)	4 (44.4%)	0.82
HF exacerbation	31 (35.2%)	2 (22.2%)	0.64
SBP on admission (mmHg)	113.2 ± 23.7	98.3 ± 10.0	0.03
DBP on admission (mmHg)	74.7 ± 12.9	66.8 ± 7.2	0.07
HR on discharge (beats per minute)	73.6 ± 13.1	75.6 ± 14.7	0.64
Comorbidities			
DM	26 (29.5%)	1 (11.1%)	0.43
COPD	8 (9.1%)	0	0.76
CKD	14 (15.9%)	1 (11.1%)	0.93
hypertension	46 (52.3%)	5 (55.6%)	0.87
NYHA class			0.84
I	3 (3.4%)	0	0.57
II	39 (44.3%)	3 (33.3%)	0.70
III	39 (44.3%)	5 (55.6%)	0.52
IV	7 (8.0%)	1 (11.1%)	0.74
I–II	42 (47.7%)	3 (33.3%)	0.64
III–IV	46 (52.3%)	6 (66.7%)	
Biochemical parameters			
Na^+^ (mmol/L)	138.9 ± 3.7	139.2 ± 2.5	0.97
BNP level (pg/mL)	685 ± 728	1038 ± 853	0.18
uric acid (µmol/L)	460.1 ± 136.1	462.1 ± 139.1	0.90
creatinine (µmol/L)	101.4 ± 29.5	105.4 ± 38.9	0.93
eGFR	73.8 ± 23.6	67.0 ± 18.7	0.64
K^+^ (mmol/L)	4.29 ± 0.39	4.04 ± 0.35	0.07
hsCRP (mg/L)	7.0 ± 9.3	5.7 ± 4.3	0.74
fasting glucose (mmol/l)	6.44 ± 2.07	5.62 ± 1.20	0.20
serum protein (g/L)	71.8 ± 7.2	66.1 ± 9.3	0.08
serum albumin (g/L)	41.1 ± 4.5	33.3 ± 5.4	<0.00001
TBIL (µmol/L)	19.3 ± 10.4	23.0 ± 17.3	0.62
ALT (U/L)	39.0 ± 22.4	32.2 ± 14.2	0.49
GGTP (U/L)	117.0 ± 122.6	95.0 ± 83.8	0.91
CholT (mmol/L)	4.33 ± 1.16	4.56 ± 0.70	0.40
TG (mmol/L)	1.55 ± 0.81	1.46 ± 0.66	0.87
LDL (mmol/L)	2.60 ± 0.92	2.70 ± 0.67	0.58
HDL (mmol/L)	1.21 ± 0.32	1.19 ± 0.34	0.75
Hb [mmol/L]	9.0 ± 1.0	8.4 ± 1.6	0.38
Nutritional parameters			
CNAQ score	28.6 ± 3.9	27.9 ± 2.9	0.47
impaired appetite CNAQ ≤ 28	41 (47%)	4 (44.4%)	0.81
MNA score	23.5 ± 2.4	21.4 ± 3.1	0.046
malnourished (MNA score < 17)	0	0	
at risk of malnutrition (MNA score 17–23.5)	44 (50%)	3 (33.3%)	0.34
normal nutritional status (MNA score > 23.5)	44 (50%)	6 (66.6%)	0.34
GNRI	115.1 ± 10.7	92.2 ± 4.4	<0.00001
Medications			
loop diuretics	82 (93.2%)	9 (100%)	0.96
thiazides	15 (17.0%)	2 (22.2%)	0.93
ß-blocker	85 (96.6%)	8 (88.9%)	0.66
ACEI/ARB	67 (76.1%)	6 (66.7%)	0.78
ARNI	14 (15.9%)	1 (11.1%)	0.94
MRA	77 (87.5%)	9 (100%)	0.62
Ca-blocker	6 (6.8%)	0	0.93
statin	54 (61.4%)	5 (55.6%)	0.98
Echocardiographic parameters			
LVEF (%)	23.4 ± 7.8	23.8 ± 7.6	0.83
LVEDD (mm)	71.3 ± 11.0	61.2 ± 11.5	0.02
RVD (mm)	37.8 ± 8.5	37.8 ± 7.6	0.96
LAD (mm)	52.9 ± 10.6	48.0 ± 8.6	0.21
IVS (mm)	10.0 ± 1.7	11.4 ± 3.4	0.37
PWD (mm)	9.9 ± 1.3	11.2 ± 4.5	0.84
Ao (mm)	33.5 ± 5.0	32.6 ± 4.1	0.49

BMI—body mass index, IHD—ischemic heart disease, SBP—systolic blood pressure, DBP—diastolic blood pressure, HR—heart rate, DM—diabetes mellitus, COPD—chronic obstructive pulmonary disease, CKD—chronic kidney disease, NYHA—New York Heart Association Classification, Na^+^ sodium concentration, BNP—B-type natriuretic peptide, NT-proBNP—N-terminal pro B-type natriuretic peptide, eGFR–estimated glomerular filtration rate, K^+^—potassium concentration, hsCRP-high-sensitivity C-reactive protein, TBIL—total bilirubin, ALT—alanine transaminase, GGTP—gamma-glutamyltranspeptidase, CholT—total cholesterol, TG—triglycerides, LDL—low density lipoprotein, HDL—high density lipoprotein, Hb—hemoglobin, CNAQ—The Council on Nutrition Appetite Questionnaire, MNA—The Mini Nutritional Assessment, GNRI—Geriatric Nutritional Risk Index, ACEI—angiotensin converting enzyme inhibitor, ARB—angiotensin receptor blocker, ARNI—angiotensin receptor-neprilysin inhibitor, MRA—mineralocorticoid receptor antagonist, LVEF—left ventricular ejection fraction, LVEDD—left ventricular end-diastolic diameter, RVD—right ventricular diameter, LAD—left atrium diameter, IVS—interventricular septum thickness, PWD—posterior wall of left ventricle, Ao—aorta.

## Data Availability

The data presented in this study are available on request from the corresponding author.

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
