# Peer review of "Appetite and Nutritional Status as Potential Management Targets in Patients with Heart Failure with Reduced Ejection Fraction—The Relationship between Echocardiographic and Biochemical Parameters and Appetite"

_jpm, 2021, doi:10.3390/jpm11070639_

Round 1
Reviewer 1 Report
Thank you for allowing me to review this manuscript titled "appetite and nutritional status as potential management targets in
patients with heart failure with reduced ejection fraction - the relationship
between echocardiographic and biochemical parameters and appetite." by Kaluzna-Oleksy et al. The manuscript discusses the relationship between nutritional status and appetite with heart failure in a prospective study using biochemical and echo parameters. It is a well written manuscript and I do not have any additional comments except a few
- Why males constitute >80% of the study population. It can create a selection bias and should be included in limitations
- Is there a figure for correlation analysis? Figure 1 is for ROC only
Author Response
Comment 1: Why males constitute >80% of the study population. It can create a selection bias and should be included in limitations Response: This situation is caused by the gender differences in prevalence of heart failure and its main cause – ischemic heart disease – it more often affects males. We included appropriate sentence in the study limitations. Comment 2: Is there a figure for correlation analysis? Figure 1 is for ROC only Response: We sincerely thank the reviewer for the precise remark. We decided not to present the figure of correlation analysis, because it provided no new important information except for those already included in the text (values of r and p). Due to our mistake the “Figure 1” was mentioned in the point “3.3 Correlation analysis” and we removed it. Properly it should occur only in the point “3.4 ROC curve analysis”.Reviewer 2 Report
1 review table 1, no need to repeat -% in the second column
2 . Table 2: needs to put p values in NHYA class and nutritional parameters
3.Discussion must be reduced
Author Response
Comment 1: ”Review table 1, no need to repeat -% in the second column”
Response: We appreciate the reviewer’s comment. We removed “%” from all the tables in columns where it was not necessary.
Comment 2: “Table 2: needs to put p values in NHYA class and nutritional parameters”
Response: According to your comment, we added p values for every single NYHA class and nutritional status. In original manuscript we had p values from Chi2 tests done for all four NYHA classes together.
Comment 3: “Discussion must be reduced”
Response: According to your comment, disscussion and bibliography were reduced and modified. The changes in the manuscript are marked up using the “Track Changes” function.